# Robust Sparse Principal Component Regression under the High Dimensional Elliptical Model

**Fang Han**
Department of Biostatistics
Johns Hopkins University
Baltimore, MD 21210
fhan@jhsph.edu

**Han Liu**
Department of Operations Research
and Financial Engineering
Princeton University
Princeton, NJ 08544
hanliu@princeton.edu

## Abstract

In this paper we focus on the principal component regression and its application to high dimension non-Gaussian data. The major contributions are two folds. First, in low dimensions and under the Gaussian model, by borrowing the strength from recent development in minimax optimal principal component estimation, we first time sharply characterize the potential advantage of classical principal component regression over least square estimation. Secondly, we propose and analyze a new robust sparse principal component regression on high dimensional elliptically distributed data. The elliptical distribution is a semiparametric generalization of the Gaussian, including many well known distributions such as multivariate Gaussian, rank-deficient Gaussian, $t$, Cauchy, and logistic. It allows the random vector to be heavy tailed and have tail dependence. These extra flexibilities make it very suitable for modeling finance and biomedical imaging data. Under the elliptical model, we prove that our method can estimate the regression coefficients in the optimal parametric rate and therefore is a good alternative to the Gaussian based methods. Experiments on synthetic and real world data are conducted to illustrate the empirical usefulness of the proposed method.

## 1 Introduction

Principal component regression (PCR) has been widely used in statistics for years (Kendall, 1968). Take the classical linear regression with random design for example. Let $\boldsymbol{x}_1, \ldots, \boldsymbol{x}_n \in \mathbb{R}^d$ be $n$ independent realizations of a random vector $\boldsymbol{X} \in \mathbb{R}^d$ with mean $\mathbf{0}$ and covariance matrix $\boldsymbol{\Sigma}$. The classical linear regression model and simple principal component regression model can be elaborated as follows:

$$\text{(Classical linear regression model)} \qquad \boldsymbol{Y} = \mathbf{X}\boldsymbol{\beta} + \boldsymbol{\epsilon};$$
$$\text{(Principal Component Regression Model)} \qquad \boldsymbol{Y} = \alpha\mathbf{X}\boldsymbol{u}_1 + \boldsymbol{\epsilon}, \qquad (1.1)$$

where $\mathbf{X} = (\boldsymbol{x}_1, \ldots, \boldsymbol{x}_n)^T \in \mathbb{R}^{n \times d}$, $\boldsymbol{Y} \in \mathbb{R}^n$, $\boldsymbol{u}_i$ is the $i$-th leading eigenvector of $\boldsymbol{\Sigma}$, and $\boldsymbol{\epsilon} \in N_n(\mathbf{0}, \sigma^2\mathbf{I}_d)$ is independent of $\mathbf{X}$, $\boldsymbol{\beta} \in \mathbb{R}^d$ and $\alpha \in \mathbb{R}$. Here $\mathbf{I}_d \in \mathbb{R}^{d \times d}$ is the identity matrix. The principal component regression then can be conducted in two steps: First we obtain an estimator $\widehat{\boldsymbol{u}}_1$ of $\boldsymbol{u}_1$; Secondly we project the data in the direction of $\widehat{\boldsymbol{u}}_1$ and solve a simple linear regression in estimating $\alpha$.

By checking Equation (1.1), it is easy to observe that the principal component regression model is a subset of the general linear regression (LR) model with the constraint that the regression coefficient $\boldsymbol{\beta}$ is proportional to $\boldsymbol{u}_1$. There has been a lot of discussions on the advantage of principal component regression over classical linear regression. In low dimensional settings, Massy (1965) pointed out that principal component regression can be much more efficient in handling collinearity among predictors compared to the linear regression. More recently, Cook (2007) and Artemiou and Li (2009) argued that principal component regression has potential to play a more important role. In particular, letting $\widehat{\boldsymbol{u}}_j$ be the $j$-th leading eigenvector of the sample covariance matrix $\widehat{\boldsymbol{\Sigma}}$ of $\boldsymbol{x}_1, \ldots, \boldsymbol{x}_n$,

Artemiou and Li (2009) show that under mild conditions with high probability the correlation between the response $\boldsymbol{Y}$ and $\mathbf{X}\widehat{\boldsymbol{u}}_i$ is higher than or equal to the correlation between $\boldsymbol{Y}$ and $\mathbf{X}\widehat{\boldsymbol{u}}_j$ when $i < j$. This indicates, although not rigorous, there is possibility that principal component regression can borrow strength from the low rank structure of $\boldsymbol{\Sigma}$, which motivates our work.

Even though the statistical performance of principal component regression in low dimensions is not fully understood, there is even less analysis on principal component regression in high dimensions where the dimension $d$ can be even exponentially larger than the sample size $n$. This is partially due to the fact that estimating the leading eigenvectors of $\boldsymbol{\Sigma}$ itself has been difficult enough. For example, Johnstone and Lu (2009) show that, even under the Gaussian model, when $d/n \to \gamma$ for some $\gamma > 0$, there exist multiple settings under which $\widehat{\boldsymbol{u}}_1$ can be an inconsistent estimator of $\boldsymbol{u}_1$. To attack this "curse of dimensionality", one solution is adding a sparsity assumption on $\boldsymbol{u}_1$, leading to various versions of the sparse PCA. See, Zou et al. (2006); d'Aspremont et al. (2007); Moghaddam et al. (2006), among others. Under the (sub)Gaussian settings, minimax optimal rates are being established in estimating $\boldsymbol{u}_1, \ldots, \boldsymbol{u}_m$ (Vu and Lei, 2012; Ma, 2013; Cai et al., 2013). Very recently, Han and Liu (2013b) relax the Gaussian assumption in conducting a scale invariant version of the sparse PCA (i.e., estimating the leading eigenvector of the correlation instead of the covariance matrix). However, it can not be easily applied to estimate $\boldsymbol{u}_1$ and the rate of convergence they proved is not the parametric rate.

This paper improves upon the aforementioned results in two directions. First, with regard to the classical principal component regression, under a double asymptotic framework in which $d$ is allowed to increase with $n$, by borrowing the very recent development in principal component analysis (Vershynin, 2010; Lounici, 2012; Bunea and Xiao, 2012), we for the first time explicitly show the advantage of principal component regression over the classical linear regression. We explicitly confirm the following two advantages of principal component regression: (i) Principal component regression is insensitive to collinearity, while linear regression is very sensitive to; (ii) Principal component regression can utilize the low rank structure of the covariance matrix $\boldsymbol{\Sigma}$, while linear regression cannot.

Secondly, in high dimensions where $d$ can increase much faster, even exponentially faster, than $n$, we propose a robust method in conducting (sparse) principal component regression under a non-Gaussian elliptical model. The elliptical distribution is a semiparametric generalization to the Gaussian, relaxing the light tail and zero tail dependence constraints, but preserving the symmetry property. We refer to Klüppelberg et al. (2007) for more details. This distribution family includes many well known distributions such as multivariate Gaussian, rank deficient Gaussian, $t$, logistic, and many others. Under the elliptical model, we exploit the result in Han and Liu (2013a), who showed that by utilizing a robust covariance matrix estimator, the multivariate Kendall's tau, we can obtain an estimator $\widetilde{\boldsymbol{u}}_1$, which can recover $\boldsymbol{u}_1$ in the optimal parametric rate as shown in Vu and Lei (2012). We then exploit $\widetilde{\boldsymbol{u}}_1$ in conducting principal component regression and show that the obtained estimator $\check{\boldsymbol{\beta}}$ can estimate $\boldsymbol{\beta}$ in the optimal $\sqrt{s \log d / n}$ rate. The optimal rate in estimating $\boldsymbol{u}_1$ and $\boldsymbol{\beta}$, combined with the discussion in the classical principal component regression, indicates that the proposed method has potential to handle high dimensional complex data and has its advantage over high dimensional linear regression methods, such as ridge regression and lasso. These theoretical results are also backed up by numerical experiments on both synthetic and real world equity data.

## 2  Classical Principal Component Regression

This section is devoted to the discussion on the advantage of classical principal component regression over the classical linear regression. We start with a brief introduction of notations. Let $\mathbf{M} = [\mathbf{M}_{ij}] \in \mathbb{R}^{d \times d}$ and $\boldsymbol{v} = (v_1, ..., v_d)^T \in \mathbb{R}^d$. We denote $\boldsymbol{v}_I$ to be the subvector of $\boldsymbol{v}$ whose entries are indexed by a set $I$. We also denote $\mathbf{M}_{I,J}$ to be the submatrix of $\mathbf{M}$ whose rows are indexed by $I$ and columns are indexed by $J$. Let $\mathbf{M}_{I*}$ and $\mathbf{M}_{*J}$ be the submatrix of $\mathbf{M}$ with rows indexed by $I$, and the submatrix of $\mathbf{M}$ with columns indexed by $J$. Let $\mathrm{supp}(\boldsymbol{v}) := \{j : v_j \neq 0\}$. For $0 < q < \infty$, we define the $\ell_0$, $\ell_q$ and $\ell_\infty$ vector norms as

$$\|\boldsymbol{v}\|_0 := \mathrm{card}(\mathrm{supp}(\boldsymbol{v})), \ \|\boldsymbol{v}\|_q := \left(\sum_{i=1}^{d} |v_i|^q\right)^{1/q} \text{ and } \|\boldsymbol{v}\|_\infty := \max_{1 \leq i \leq d} |v_i|.$$

Let $\mathrm{Tr}(\mathbf{M})$ be the trace of $\mathbf{M}$. Let $\lambda_j(\mathbf{M})$ be the $j$-th largest eigenvalue of $\mathbf{M}$ and $\boldsymbol{\Theta}_j(\mathbf{M})$ be the corresponding leading eigenvector. In particular, we let $\lambda_{\max}(\mathbf{M}) := \lambda_1(\mathbf{M})$ and $\lambda_{\min}(\mathbf{M}) :=$

$\lambda_d(\mathbf{M})$. We define $\mathbb{S}^{d-1} := \{\boldsymbol{v} \in \mathbb{R}^d : \|\boldsymbol{v}\|_2 = 1\}$ to be the $d$-dimensional unit sphere. We define the matrix $\ell_{\max}$ norm and $\ell_2$ norm as $\|\mathbf{M}\|_{\max} := \max\{|\mathbf{M}_{ij}|\}$ and $\|\mathbf{M}\|_2 := \sup_{\boldsymbol{v} \in \mathbb{S}^{d-1}} \|\mathbf{M}\boldsymbol{v}\|_2$. We define $\operatorname{diag}(\mathbf{M})$ to be a diagonal matrix with $[\operatorname{diag}(\mathbf{M})]_{jj} = \mathbf{M}_{jj}$ for $j = 1, \ldots, d$. We denote $\operatorname{vec}(\mathbf{M}) := (\mathbf{M}_{*1}^T, \ldots, \mathbf{M}_{*d}^T)^T$. For any two sequence $\{a_n\}$ and $\{b_n\}$, we denote $a_n \overset{c,C}{\asymp} b_n$ if there exist two fixed constants $c, C$ such that $c \leq a_n/b_n \leq C$.

Let $\boldsymbol{x}_1, \ldots, \boldsymbol{x}_n \in \mathbb{R}^d$ be $n$ independent observations of a $d$-dimensional random vector $\boldsymbol{X} \sim N_d(\mathbf{0}, \boldsymbol{\Sigma})$, $\boldsymbol{u}_1 := \boldsymbol{\Theta}_1(\boldsymbol{\Sigma})$ and $\epsilon_1, \ldots, \epsilon_n \sim N_1(0, \sigma^2)$ are independent from each other and $\{\boldsymbol{X}_i\}_{i=1}^n$. We suppose that the following principal component regression model holds:

$$\boldsymbol{Y} = \alpha \mathbf{X} \boldsymbol{u}_1 + \boldsymbol{\epsilon}, \tag{2.1}$$

where $\boldsymbol{Y} = (Y_1, \ldots, Y_n)^T \in \mathbb{R}^n$, $\mathbf{X} = [\boldsymbol{x}_1, \ldots, \boldsymbol{x}_n]^T \in \mathbb{R}^{n \times d}$ and $\boldsymbol{\epsilon} = (\epsilon_1, \ldots, \epsilon_n)^T \in \mathbb{R}^n$. We are interested in estimating the regression coefficient $\boldsymbol{\beta} := \alpha \boldsymbol{u}_1$.

Let $\widehat{\boldsymbol{\beta}}$ represent the solution of the classical least square estimator without taking the information that $\boldsymbol{\beta}$ is proportional to $\boldsymbol{u}_1$ into account. $\widehat{\boldsymbol{\beta}}$ can be expressed as follows:

$$\widehat{\boldsymbol{\beta}} := (\mathbf{X}^T \mathbf{X})^{-1} \mathbf{X}^T \boldsymbol{Y}. \tag{2.2}$$

We then have the following proposition, which shows that the mean square error of $\widehat{\boldsymbol{\beta}} - \boldsymbol{\beta}$ is highly related to the scale of $\lambda_{\min}(\boldsymbol{\Sigma})$.

**Proposition 2.1.** *Under the principal component regression model shown in* (2.1)*, we have*

$$\mathbb{E}\|\widehat{\boldsymbol{\beta}} - \boldsymbol{\beta}\|_2^2 = \frac{\sigma^2}{n - d - 1} \left( \frac{1}{\lambda_1(\boldsymbol{\Sigma})} + \cdots + \frac{1}{\lambda_d(\boldsymbol{\Sigma})} \right).$$

Proposition 2.1 reflects the vulnerability of least square estimator on the collinearity. More specifically, when $\lambda_d(\boldsymbol{\Sigma})$ is extremely small, going to zero in the scale of $O(1/n)$, $\widehat{\boldsymbol{\beta}}$ can be an inconsistent estimator even when $d$ is fixed. On the other hand, using the Markov inequality, when $\lambda_d(\boldsymbol{\Sigma})$ is lower bounded by a fixed constant and $d = o(n)$, the rate of convergence of $\widehat{\boldsymbol{\beta}}$ is well known to be $O_P(\sqrt{d/n})$.

Motivated from Equation (2.1), the classical principal component regression estimator can be elaborated as follows.

(1) We first estimate $\boldsymbol{u}_1$ using the leading eigenvector $\widehat{\boldsymbol{u}}_1$ of the sample covariance $\widehat{\boldsymbol{\Sigma}} := \frac{1}{n} \sum \boldsymbol{x}_i \boldsymbol{x}_i^T$.

(2) We then estimate $\alpha \in \mathbb{R}$ in Equation (2.1) by the standard least square estimation on the projected data $\widehat{\boldsymbol{Z}} := \mathbf{X} \widehat{\boldsymbol{u}}_1 \in \mathbb{R}^n$:

$$\widetilde{\alpha} := (\widehat{\boldsymbol{Z}}^T \widehat{\boldsymbol{Z}})^{-1} \widehat{\boldsymbol{Z}}^T \boldsymbol{Y},$$

The final principal component regression estimator $\widetilde{\boldsymbol{\beta}}$ is then obtained as $\widetilde{\boldsymbol{\beta}} = \widetilde{\alpha} \widehat{\boldsymbol{u}}_1$. We then have the following important theorem, which provides a rate of convergence for $\widetilde{\boldsymbol{\beta}}$ to approximate $\boldsymbol{\beta}$.

**Theorem 2.2.** *Let $r^*(\boldsymbol{\Sigma}) := \operatorname{Tr}(\boldsymbol{\Sigma})/\lambda_{\max}(\boldsymbol{\Sigma})$ represent the effective rank of $\boldsymbol{\Sigma}$ (Vershynin, 2010). Suppose that*

$$\|\boldsymbol{\Sigma}\|_2 \cdot \sqrt{\frac{r^*(\boldsymbol{\Sigma}) \log d}{n}} = o(1).$$

*Under the Model* (2.1)*, when $\lambda_{\max}(\boldsymbol{\Sigma}) > c_1$ and $\lambda_2(\boldsymbol{\Sigma})/\lambda_1(\boldsymbol{\Sigma}) < C_1 < 1$ for some fixed constants $C_1$ and $c_1$, we have*

$$\|\widetilde{\boldsymbol{\beta}} - \boldsymbol{\beta}\|_2 = O_P \left\{ \sqrt{\frac{1}{n}} + \left( \alpha + \frac{1}{\sqrt{\lambda_{\max}(\boldsymbol{\Sigma})}} \right) \cdot \sqrt{\frac{r^*(\boldsymbol{\Sigma}) \log d}{n}} \right\}. \tag{2.3}$$

Theorem 2.2, compared to Proposition 2.1, provides several important messages on the performance of principal component regression. First, compared to the least square estimator $\widehat{\boldsymbol{\beta}}$, $\widetilde{\boldsymbol{\beta}}$ is insensitive to collinearity in the sense that $\lambda_{\min}(\boldsymbol{\Sigma})$ plays no role in the rate of convergence of $\widetilde{\boldsymbol{\beta}}$. Secondly, when $\lambda_{\min}(\boldsymbol{\Sigma})$ is lower bounded by a fixed constant and $\alpha$ is upper bounded by a fixed constant, the rate of convergence for $\widehat{\boldsymbol{\beta}}$ is $O_P(\sqrt{d/n})$ and for $\widetilde{\boldsymbol{\beta}}$ is $O_P(\sqrt{r^*(\boldsymbol{\Sigma}) \log d/n})$, while $r^*(\boldsymbol{\Sigma}) :=$

$\operatorname{Tr}(\boldsymbol{\Sigma})/\lambda_{\max}(\boldsymbol{\Sigma}) \leq d$ and is of order $o(d)$ when there exists a low rank structure for $\boldsymbol{\Sigma}$. These two observations, combined together, illustrate the advantages of the classical principal component regression over least square estimation. These advantages justify the use of principal component regression. There is one more thing to be noted: the performance of $\widetilde{\boldsymbol{\beta}}$, unlike $\widehat{\boldsymbol{\beta}}$, depends on $\alpha$. When $\alpha$ is small, $\widetilde{\boldsymbol{\beta}}$ can predict $\boldsymbol{\beta}$ more accurately.

These three observations are verified in Figure 1. Here the data are generated according to Equation (2.1) and we set $n = 100$, $d = 10$, $\boldsymbol{\Sigma}$ to be a diagonal matrix with descending diagonal values $\boldsymbol{\Sigma}_{ii} = \lambda_i$ and $\sigma^2 = 1$. In Figure 1(A), we set $\alpha = 1$, $\lambda_1 = 10$, $\lambda_j = 1$ for $j = 2, \ldots, d-1$, and changing $\lambda_d$ from 1 to 1/100; In Figure 1(B), we set $\alpha = 1$, $\lambda_j = 1$ for $j = 2, \ldots, d$ and changing $\lambda_1$ from 1 to 100; In Figure 1(C), we set $\lambda_1 = 10$, $\lambda_j = 1$ for $j = 2, \ldots, d$, and changing $\alpha$ from 0.1 to 10. In the three figures, the empirical mean square error is plotted against $1/\lambda_d$, $\lambda_1$, and $\alpha$. It can be observed that the results, each by each, matches the theory.

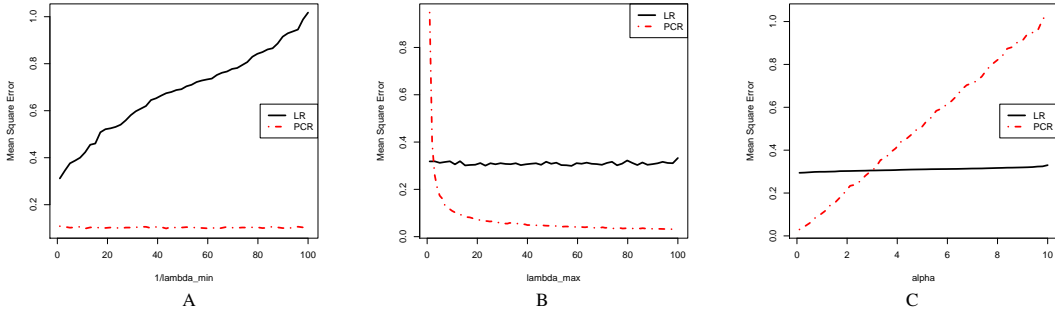

Figure 1: Justification of Proposition 2.1 and Theorem 2.2. The empirical mean square errors are plotted against $1/\lambda_d$, $\lambda_1$, and $\alpha$ separately in (A), (B), and (C). Here the results of classical linear regression and principal component regression are marked in black solid line and red dotted line.

## 3 Robust Sparse Principal Component Regression under Elliptical Model

In this section, we propose a new principal component regression method. We generalize the settings in classical principal component regression discussed in the last section in two directions: (i) We consider the high dimensional settings where the dimension $d$ can be much larger than the sample size $n$; (ii) In modeling the predictors $\boldsymbol{x}_1, \ldots, \boldsymbol{x}_n$, we consider a more general elliptical, instead of the Gaussian distribution family. The elliptical family can capture characteristics such as heavy tails and tail dependence, making it more suitable for analyzing complex datasets in finance, genomics, and biomedical imaging.

### 3.1 Elliptical Distribution

In this section we define the elliptical distribution and introduce the basic property of the elliptical distribution. We denote by $\boldsymbol{X} \overset{d}{=} \boldsymbol{Y}$ if random vectors $\boldsymbol{X}$ and $\boldsymbol{Y}$ have the same distribution.

Here we only consider the continuous random vectors with density existing. To our knowledge, there are essentially four ways to define the continuous elliptical distribution with density. The most intuitive way is as follows: A random vector $\boldsymbol{X} \in \mathbb{R}^d$ is said to follow an elliptical distribution $EC_d(\boldsymbol{\mu}, \boldsymbol{\Sigma}, \xi)$ if and only there exists a random variable $\xi > 0$ ($a.s.$) and a Gaussian distribution $\boldsymbol{Z} \sim N_d(\boldsymbol{0}, \boldsymbol{\Sigma})$ such that

$$\boldsymbol{X} \overset{d}{=} \boldsymbol{\mu} + \xi \boldsymbol{Z}. \tag{3.1}$$

Note that here $\xi$ is not necessarily independent of $\boldsymbol{Z}$. Accordingly, elliptical distribution can be regarded as a semiparametric generalization to the Gaussian distribution, with the nonparametric part $\xi$. Because $\xi$ can be very heavy tailed, $\boldsymbol{X}$ can also be very heavy tailed. Moreover, when $\mathbb{E}\xi^2$ exists, we have

$$\operatorname{Cov}(\boldsymbol{X}) = \mathbb{E}\xi^2 \boldsymbol{\Sigma} \quad \text{and} \quad \boldsymbol{\Theta}_j(\operatorname{Cov}(\boldsymbol{X})) = \boldsymbol{\Theta}_j(\boldsymbol{\Sigma}) \quad \text{for } j = 1, \ldots, d.$$

This implies that, when $\mathbb{E}\xi^2$ exists, to recover $\boldsymbol{u}_1 := \boldsymbol{\Theta}_1(\operatorname{Cov}(\boldsymbol{X}))$, we only need to recover $\boldsymbol{\Theta}_1(\boldsymbol{\Sigma})$. Here $\boldsymbol{\Sigma}$ is conventionally called the scatter matrix.

We would like to point out that the elliptical family is significantly larger than the Gaussian. In fact, Gaussian is fully parameterized by finite dimensional parameters (mean and variance). In contrast, the elliptical is a semiparametric family (since the elliptical density can be represented as $g((\boldsymbol{x}-\boldsymbol{\mu})^T \boldsymbol{\Sigma}^{-}1(\boldsymbol{x}-\boldsymbol{\mu}))$ where the function $g(\cdot)$ function is completely unspecified.). If we consider the "volumes" of the family of the elliptical family and the Gaussian family with respect to the Lebesgue reference measure, the volume of Gaussian family is zero (like a line in a 3-dimensional space), while the volume of the elliptical family is positive (like a ball in a 3-dimensional space).

## 3.2 Multivariate Kendall's tau

As a important step in conducting the principal component regression, we need to estimate $\boldsymbol{u}_1 = \boldsymbol{\Theta}_1(\mathrm{Cov}(\boldsymbol{X})) = \boldsymbol{\Theta}_1(\boldsymbol{\Sigma})$ as accurately as possible. Since the random variable $\xi$ in Equation (3.1) can be very heavy tailed, the according elliptical distributed random vector can be heavy tailed. Therefore, as has been pointed out by various authors (Tyler, 1987; Croux et al., 2002; Han and Liu, 2013b), the leading eigenvector of the sample covariance matrix $\widehat{\boldsymbol{\Sigma}}$ can be a bad estimator in estimating $\boldsymbol{u}_1 = \boldsymbol{\Theta}_1(\boldsymbol{\Sigma})$ under the elliptical distribution. This motivates developing robust estimator.

In particular, in this paper we consider using the multivariate Kendall's tau (Choi and Marden, 1998) and recently deeply studied by Han and Liu (2013a). In the following we give a brief description of this estimator. Let $\boldsymbol{X} \sim EC_d(\boldsymbol{\mu}, \boldsymbol{\Sigma}, \xi)$ and $\widetilde{\boldsymbol{X}}$ be an independent copy of $\boldsymbol{X}$. The population multivariate Kendall's tau matrix, denoted by $\mathbf{K} \in \mathbb{R}^{d \times d}$, is defined as:

$$\mathbf{K} := \mathbb{E}\left( \frac{(\boldsymbol{X} - \widetilde{\boldsymbol{X}})(\boldsymbol{X} - \widetilde{\boldsymbol{X}})^T}{\|\boldsymbol{X} - \widetilde{\boldsymbol{X}}\|_2^2} \right). \tag{3.2}$$

Let $\boldsymbol{x}_1, \ldots, \boldsymbol{x}_n$ be $n$ independent observations of $\boldsymbol{X}$. The sample version of multivariate Kendall's tau is accordingly defined as

$$\widehat{\mathbf{K}} = \frac{1}{n(n-1)} \sum_{i \neq j} \frac{(\boldsymbol{x}_i - \boldsymbol{x}_j)(\boldsymbol{x}_i - \boldsymbol{x}_j)^T}{\|\boldsymbol{x}_i - \boldsymbol{x}_j\|_2^2}, \tag{3.3}$$

and we have that $\mathbb{E}(\widehat{\mathbf{K}}) = \mathbf{K}$. $\widehat{\mathbf{K}}$ is a matrix version U statistic and it is easy to see that $\max_{jk} |\mathbf{K}_{jk}| \leq 1, \max_{jk} |\widehat{\mathbf{K}}_{jk}| \leq 1$. Therefore, $\widehat{\mathbf{K}}$ is a bounded matrix and hence can be a nicer statistic than the sample covariance matrix. Moreover, we have the following important proposition, coming from Oja (2010), showing that $\mathbf{K}$ has the same eigenspace as $\boldsymbol{\Sigma}$ and $\mathrm{Cov}(\boldsymbol{X})$.

**Proposition 3.1** (Oja (2010))**.** *Let $\boldsymbol{X} \sim EC_d(\boldsymbol{\mu}, \boldsymbol{\Sigma}, \xi)$ be a continuous distribution and $\mathbf{K}$ be the population multivariate Kendall's tau statistic. Then if $\lambda_j(\boldsymbol{\Sigma}) \neq \lambda_k(\boldsymbol{\Sigma})$ for any $k \neq j$, we have*

$$\boldsymbol{\Theta}_j(\boldsymbol{\Sigma}) = \boldsymbol{\Theta}_j(\mathbf{K}) \quad \text{and} \quad \lambda_j(\mathbf{K}) = \mathbb{E}\left( \frac{\lambda_j(\boldsymbol{\Sigma})U_j^2}{\lambda_1(\boldsymbol{\Sigma})U_1^2 + \ldots + \lambda_d(\boldsymbol{\Sigma})U_d^2} \right), \tag{3.4}$$

*where $\boldsymbol{U} := (U_1, \ldots, U_d)^T$ follows a uniform distribution in $\mathbb{S}^{d-1}$. In particular, when $\mathbb{E}\xi^2$ exists, $\boldsymbol{\Theta}_j(\mathrm{Cov}(\boldsymbol{X})) = \boldsymbol{\Theta}_j(\mathbf{K})$.*

## 3.3 Model and Method

In this section we discuss the model we build and the accordingly proposed method in conducting high dimensional (sparse) principal component regression on non-Gaussian data.

Similar as in Section 2, we consider the classical simple principal component regression model:

$$\boldsymbol{Y} = \alpha \mathbf{X} \boldsymbol{u}_1 + \boldsymbol{\epsilon} = \alpha [\boldsymbol{x}_1, \ldots, \boldsymbol{x}_n]^T \boldsymbol{u}_1 + \boldsymbol{\epsilon}.$$

To relax the Gaussian assumption, we assume that both $\boldsymbol{x}_1, \ldots, \boldsymbol{x}_n \in \mathbb{R}_d$ and $\epsilon_1, \ldots, \epsilon_n \in \mathbb{R}$ be elliptically distributed. We assume that $\boldsymbol{x}_i \in EC_d(\boldsymbol{0}, \boldsymbol{\Sigma}, \xi)$. To allow the dimension $d$ increasing much faster than $n$, we impose a sparsity structure on $\boldsymbol{u}_1 = \boldsymbol{\Theta}_1(\boldsymbol{\Sigma})$. Moreover, to make $\boldsymbol{u}_1$ identifiable, we assume that $\lambda_1(\boldsymbol{\Sigma}) \neq \lambda_2(\boldsymbol{\Sigma})$. Thusly, the formal model of the robust sparse principal component regression considered in this paper is as follows:

$$\mathcal{M}_d(\boldsymbol{Y}, \boldsymbol{\epsilon}; \boldsymbol{\Sigma}, \xi, s): \quad \begin{cases} \boldsymbol{Y} = \alpha \mathbf{X} \boldsymbol{u}_1 + \boldsymbol{\epsilon}, \\ \boldsymbol{x}_1, \ldots, \boldsymbol{x}_n \sim EC_d(\boldsymbol{0}, \boldsymbol{\Sigma}, \xi), \quad \|\boldsymbol{\Theta}_1(\boldsymbol{\Sigma})\|_0 \leq s, \lambda_1(\boldsymbol{\Sigma}) \neq \lambda_2(\boldsymbol{\Sigma}). \end{cases} \tag{3.5}$$

Then the robust sparse principal component regression can be elaborated as a two step procedure:

(i) Inspired by the model $\mathcal{M}_d(\boldsymbol{Y}, \boldsymbol{\epsilon}; \boldsymbol{\Sigma}, \xi, s)$ and Proposition 3.1, we consider the following optimization problem to estimate $\boldsymbol{u}_1 := \boldsymbol{\Theta}_1(\boldsymbol{\Sigma})$:

$$\widetilde{\boldsymbol{u}}_1 = \arg\max_{\boldsymbol{v} \in \mathbb{R}^d} \boldsymbol{v}^T \widehat{\boldsymbol{K}} \boldsymbol{v}, \quad \text{subject to} \quad \boldsymbol{v} \in \mathbb{S}^{d-1} \cap \mathbb{B}_0(s), \tag{3.6}$$

where $\mathbb{B}_0(s) := \{\boldsymbol{v} \in \mathbb{R}^d : \|\boldsymbol{v}\|_0 \leq s\}$ and $\widehat{\boldsymbol{K}}$ is the estimated multivariate Kendall's tau matrix. The corresponding global optimum is denoted by $\widetilde{\boldsymbol{u}}_1$. Using Proposition 3.1, $\widetilde{\boldsymbol{u}}_1$ is also an estimator of $\boldsymbol{\Theta}_1(\mathrm{Cov}(\boldsymbol{X}))$, whenever the covariance matrix exists.

(ii) We then estimate $\alpha \in \mathbb{R}$ in Equation (3.5) by the standard least square estimation on the projected data $\widetilde{\boldsymbol{Z}} := \boldsymbol{X}\widetilde{\boldsymbol{u}}_1 \in \mathbb{R}^n$:

$$\check{\alpha} := (\widetilde{\boldsymbol{Z}}^T \widetilde{\boldsymbol{Z}})^{-1} \widetilde{\boldsymbol{Z}}^T \boldsymbol{Y},$$

The final principal component regression estimator $\check{\boldsymbol{\beta}}$ is then obtained as $\check{\boldsymbol{\beta}} = \check{\alpha}\widetilde{\boldsymbol{u}}_1$.

### 3.4 Theoretical Property

In Theorem 2.2, we show that how to estimate $\boldsymbol{u}_1$ accurately plays an important role in conducting the principal component regression. Following this discussion and the very recent results in Han and Liu (2013a), the following "easiest" and "hardest" conditions are considered. Here $\kappa_L, \kappa_U$ are two constants larger than 1.

Condition 1 ("Easiest"): $\lambda_1(\boldsymbol{\Sigma}) \overset{1,\kappa_U}{\asymp} d\lambda_j(\boldsymbol{\Sigma})$ for any $j \in \{2, \ldots, d\}$ and $\lambda_2(\boldsymbol{\Sigma}) \overset{1,\kappa_U}{\asymp} \lambda_j(\boldsymbol{\Sigma})$ for any $j \in \{3, \ldots, d\}$;

Condition 2 ("Hardest"): $\lambda_1(\boldsymbol{\Sigma}) \overset{\kappa_L,\kappa_U}{\asymp} \lambda_j(\boldsymbol{\Sigma})$ for any $j \in \{2, \ldots, d\}$.

In the sequel, we say that the model $\mathcal{M}_d(\boldsymbol{Y}, \boldsymbol{\epsilon}; \boldsymbol{\Sigma}, \xi, s)$ holds if the data $(\boldsymbol{Y}, \boldsymbol{X})$ are generated using the model $\mathcal{M}_d(\boldsymbol{Y}, \boldsymbol{\epsilon}; \boldsymbol{\Sigma}, \xi, s)$.

Under Conditions 1 and 2, we then have the following theorem, which shows that under certain conditions, $\|\check{\boldsymbol{\beta}} - \boldsymbol{\beta}\|_2 = O_P(\sqrt{s \log d/n})$, which is the optimal parametric rate in estimating the regression coefficient (Ravikumar et al., 2008).

**Theorem 3.2.** *Let the model $\mathcal{M}_d(\boldsymbol{Y}, \boldsymbol{\epsilon}; \boldsymbol{\Sigma}, \xi, s)$ hold and $|\alpha|$ in Equation* (3.5) *are upper bounded by a constant and $\|\boldsymbol{\Sigma}\|_2$ is lower bounded by a constant. Then under Condition 1 or Condition 2 and for all random vector $\boldsymbol{X}$ such that*

$$\max_{\boldsymbol{v} \in \mathbb{S}^{d-1}, \|\boldsymbol{v}\|_0 \leq 2s} |\boldsymbol{v}^T(\widehat{\boldsymbol{\Sigma}} - \boldsymbol{\Sigma})\boldsymbol{v}| = o_P(1),$$

*we have the robust principal component regression estimator $\check{\boldsymbol{\beta}}$ satisfies that*

$$\|\check{\boldsymbol{\beta}} - \boldsymbol{\beta}\|_2 = O_P\left(\sqrt{\frac{s \log d}{n}}\right).$$

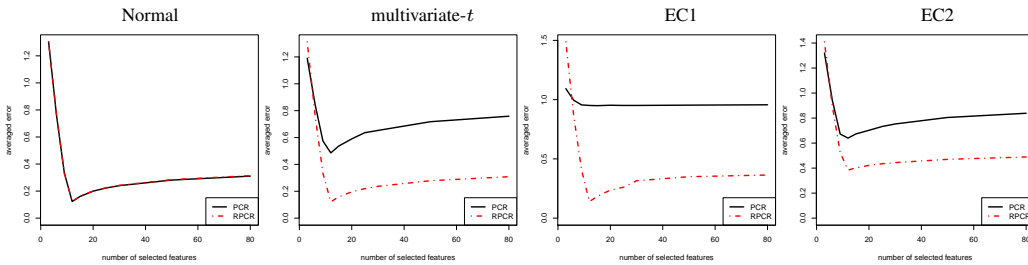

Figure 2: Curves of averaged estimation errors between the estimates and true parameters for different distributions (normal, multivariate-$t$, EC1, and EC2, from left to right) using the truncated power method. Here $n = 100$, $d = 200$, and we are interested in estimating the regression coefficient $\boldsymbol{\beta}$. The horizontal-axis represents the cardinalities of the estimates' support sets and the vertical-axis represents the empirical mean square error. Here from the left to the right, the minimum mean square errors for lasso are 0.53, 0.55, 1, and 1.

# 4 Experiments

In this section we conduct study on both synthetic and real-world data to investigate the empirical performance of the robust sparse principal component regression proposed in this paper. We use the truncated power algorithm proposed in Yuan and Zhang (2013) to approximate the global optimums $\tilde{u}_1$ to (3.6). Here the cardinalities of the support sets of the leading eigenvectors are treated as tuning parameters. The following three methods are considered:

lasso: the classical $L_1$ penalized regression;

PCR: The sparse principal component regression using the sample covariance matrix as the sufficient statistic and exploiting the truncated power algorithm in estimating $u_1$;

RPCR: The robust sparse principal component regression proposed in this paper, using the multivariate Kendall's tau as the sufficient statistic and exploiting the truncated power algorithm to estimate $u_1$.

## 4.1 Simulation Study

In this section, we conduct simulation study to back up the theoretical results and further investigate the empirical performance of the proposed robust sparse principal component regression method.

To illustrate the empirical usefulness of the proposed method, we first consider generating the data matrix $\mathbf{X}$. To generate $\mathbf{X}$, we need to consider how to generate $\boldsymbol{\Sigma}$ and $\xi$. In detail, let $\omega_1 > \omega_2 > \omega_3 = \ldots = \omega_d$ be the eigenvalues and $u_1, \ldots, u_d$ be the eigenvectors of $\boldsymbol{\Sigma}$ with $u_j := (u_{j1}, \ldots, u_{jd})^T$. The top 2 leading eigenvectors $u_1, u_2$ of $\boldsymbol{\Sigma}$ are specified to be sparse with $s_j := \|u_j\|_0$ and $u_{jk} = 1/\sqrt{s_j}$ for $k \in [1 + \sum_{i=1}^{j-1} s_i, \sum_{i=1}^{j} s_i]$ and zero for all the others. $\boldsymbol{\Sigma}$ is generated as $\boldsymbol{\Sigma} = \sum_{j=1}^{2} (\omega_j - \omega_d) u_j u_j^T + \omega_d \mathbf{I}_d$. Across all settings, we let $s_1 = s_2 = 10, \omega_1 = 5.5, \omega_2 = 2.5$, and $\omega_j = 0.5$ for all $j = 3, \ldots, d$. With $\boldsymbol{\Sigma}$, we then consider the following four different elliptical distributions:

(**Normal**) $X \sim EC_d(\mathbf{0}, \boldsymbol{\Sigma}, \zeta_1)$ with $\zeta_1 \stackrel{d}{=} \chi_d$. Here $\chi_d$ is the chi-distribution with degree of freedom $d$. For $Y_1, \ldots, Y_d \stackrel{i.i.d.}{\sim} N(0,1)$, $\sqrt{Y_1^2 + \ldots + Y_d^2} \stackrel{d}{=} \chi_d$. In this setting, $X$ follows the Gaussian distribution (Fang et al., 1990).

(**Multivariate-$t$**) $X \sim EC_d(\mathbf{0}, \boldsymbol{\Sigma}, \zeta_2)$ with $\zeta_2 \stackrel{d}{=} \sqrt{\kappa}\xi_1^*/\xi_2^*$. Here $\xi_1^* \stackrel{d}{=} \chi_d$ and $\xi_2^* \stackrel{d}{=} \chi_\kappa$ with $\kappa \in \mathbb{Z}^+$. In this setting, $X$ follows a multivariate-$t$ distribution with degree of freedom $\kappa$ (Fang et al., 1990). Here we consider $\kappa = 3$.

(**EC1**) $X \sim EC_d(\mathbf{0}, \boldsymbol{\Sigma}, \zeta_3)$ with $\zeta_3 \sim F(d, 1)$, an $F$ distribution.

(**EC2**) $X \sim EC_d(\mathbf{0}, \boldsymbol{\Sigma}, \zeta_4)$ with $\zeta_4 \sim \text{Exp}(1)$, an exponential distribution.

We then simulate $x_1, \ldots, x_n$ from $X$. This forms a data matrix $\mathbf{X}$. Secondly, we let $Y = \mathbf{X}u_1 + \epsilon$, where $\epsilon \sim N_n(\mathbf{0}, \mathbf{I}_n)$. This produces the data $(Y, \mathbf{X})$. We repeatedly generate $n$ data according to the four distributions discussed above for 1,000 times. To show the estimation accuracy, Figure 2 plots the empirical mean square error between the estimate $\tilde{u}_1$ and true regression coefficient $\beta$ against the numbers of estimated nonzero entries (defined as $\|\tilde{u}_1\|_0$), for PCR and RPCR, under different schemes of $(n, d)$, $\boldsymbol{\Sigma}$ and different distributions. Here we considered $n = 100$ and $d = 200$.

It can be seen that we do not plot the results of lasso in Figure 2. As discussed in Section 2, especially as shown in Figure 1, linear regression and principal component regression have their own advantages in different settings. More specifically, we do not plot the results of lasso here simply because it performs so bad under our simulation settings. For example, under the Gaussian settings with $n = 100$ and $d = 200$, the lowest mean square error for lasso is 0.53 and the errors are averagely above 1.5, while for RPCR is 0.13 and is averagely below 1.

Figure 2 shows when the data are non-Gaussian but follow an elliptically distribution, RPCR outperforms PCR constantly in terms of estimation accuracy. Moreover, when the data are indeed normally distributed, there is no obvious difference between RPCR and PCR, indicating that RPCR is a safe alternative to the classical sparse principal component regression.

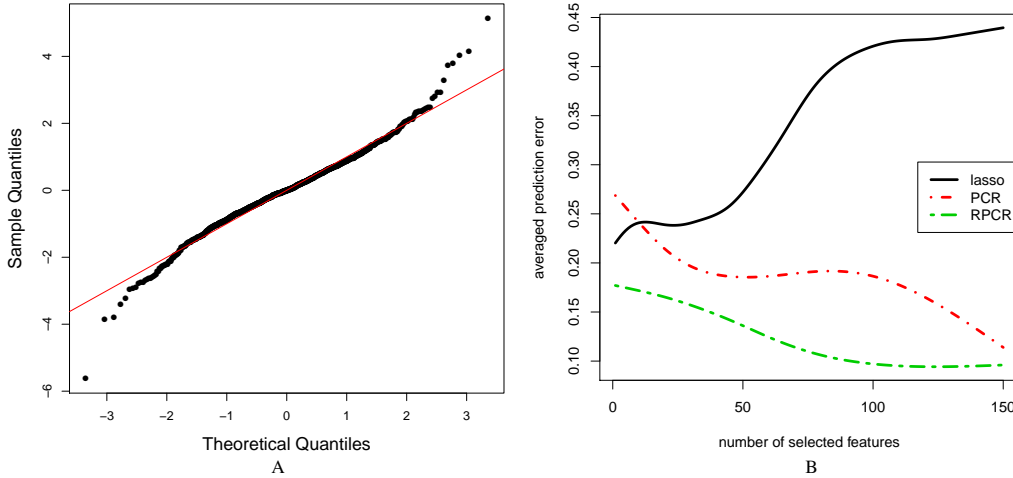

Figure 3: (A) Quantile vs. quantile plot of the log-return values for one stock "Goldman Sachs". (B) Prediction error against the number of features selected. The scale of the prediction errors is enlarged by 100 times for better visualization.

## 4.2   Application to Equity Data

In this section we apply the proposed robust sparse principal component regression and the other two methods to the stock price data from Yahoo! Finance (finance.yahoo.com). We collect the daily closing prices for 452 stocks that are consistently in the S&P 500 index between January 1, 2003 through January 1, 2008. This gives us altogether T=1,257 data points, each data point corresponds to the vector of closing prices on a trading day. Let $St = [St_{t,j}]$ denote by the closing price of stock $j$ on day $t$. We are interested in the log return data $\mathbf{X} = [\mathbf{X}_{tj}]$ with $\mathbf{X}_{tj} = \log(St_{t,j}/St_{t-1,j})$.

We first show that this data set is non-Gaussian and heavy tailed. This is done first by conducting marginal normality tests (Kolmogorove-Smirnov, Shapiro-Wilk, and Lillifors) on the data. We find that at most 24 out of 452 stocks would pass any of three normality test. With Bonferroni correction there are still over half stocks that fail to pass any normality tests. Moreover, to illustrate the heavy tailed issue, we plot the quantile vs. quantile plot for one stock, "Goldman Sachs", in Figure 3(A). It can be observed that the log return values for this stock is heavy tailed compared to the Gaussian.

To illustrate the power of the proposed method, we pick a subset of the data first. The stocks can be summarized into 10 Global Industry Classification Standard (GICS) sectors and we are focusing on the subcategory "Financial". This leave us 74 stocks and we denote the resulting data to be $\mathbf{F} \in \mathbb{R}^{1257 \times 74}$. We are interested in predicting the log return value in day $t$ for each stock indexed by $k$ (i.e., treating $\mathbf{F}_{t,k}$ as the response) using the log return values for all the stocks in day $t-1$ to day $t-7$ (i.e., treating $\text{vec}(\mathbf{F}_{t-7 \leq t' \leq t-1,\cdot})$ as the predictor). The dimension for the regressor is accordingly $7 \times 74 = 518$. For each stock indexed by $k$, to learn the regression coefficient $\boldsymbol{\beta}_k$, we use $\mathbf{F}_{t' \in \{1,...,1256\},\cdot}$ as the training data and applying the three different methods on this dataset. For each method, after obtaining an estimator $\widehat{\boldsymbol{\beta}}_k$, we use $\text{vec}(\mathbf{F}_{t' \in \{1250,...,1256\},\cdot})\widehat{\boldsymbol{\beta}}$ to estimate $\mathbf{F}_{1257,k}$. This procedure is repeated for each $k$ and the averaged prediction errors are plotted against the number of features selected (i.e., $\|\widehat{\boldsymbol{\beta}}\|_0$) in Figure 3(B). To visualize the difference more clearly, in the figures we enlarge the scale of the prediction errors by 100 times. It can be observed that RPCR has the universally lowest prediction error with regard to different number of features.

## Acknowledgement

Han's research is supported by a Google fellowship. Liu is supported by NSF Grants III-1116730 and NSF III-1332109, an NIH sub-award and a FDA sub-award from Johns Hopkins University.

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
