[Reviews · NeurIPS 2013]

Submitted by Assigned_Reviewer_5

The authors present a new method for robust principal component regression for
non-Gaussian data. First, they show that principal component regression
outperforms classical linear regression when the dimensionality and the sample
size are allowed to increase by being insensitive to collinearity and
exploiting low rank structure. They demonstrate their theoretical calculations
by sweeping parameters and show that mean square error follows theory. Then the
authors develop a new method for doing principal component regression by
assuming the random vector and noise are elliptically distributed, a more
general assumption than the standard Gaussian assumption. They demonstrate
that this more general method outperforms traditional principal component
regression on different elliptical distributions (multivariate-t, EC1, EC2),
and show that it achieves similar performance for Gaussian distributions.
Finally, they compare performance on real finance data and demonstrate that
their new method outperforms the standard principal component regression and
the standard lasso (linear regression) technique.

This paper is very high quality. The introduction presents a clear explanation
of related work and goes on to explain the significant contributions made by
this work. The sections are logically organized, and the math is explained
well. The figures support the arguments put forth by the authors. The authors
new principal component method outperforms standard principal component method
on both generated data and real world data.

The authors could clarify how they implemented lasso regression when performing
the simulation study and when analyzing equity data. How was the number of
selected features chosen from the lasso method? Was the threshold varied up and
down to change the sparsity pattern or was the lasso trade-off parameter
varied? After the features were chosen was the solution polished? That is, the
sparsity pattern can be determined from using lasso regression, but then the
regression can be re-run (polished) with the fixed sparsity pattern without the
additional $l_1$ cost function.

Finally, when looking at equity data. The authors chose a subset of all stock
data. Were other categories tested or was there a particular reason why the
authors focused on this category? Their results would be even stronger if they
demonstrated improved performance in multiple sectors.
Summary: The authors present a high quality, thorough paper on a new method for robust
principal component regression. The authors could clarify a few minor points,
but the paper is overall solid work.

Submitted by Assigned_Reviewer_7

The authors propose a robust and sparse principal component regression (PCR) estimator for non-Gaussian data. This is motivated by theoretical arguments on when classical PCR is justified over least squares regression (when a low-rank structure is present) and by data / noise with heavy and dependent tails. Finally, the approach is demonstrated successfully on simulated and experimental equity data.

The writing is very clear. There are two significant contributions:
1. The authors show the when PCR is preferable to standard least squares regression (collinearity invariance, exploitation of low-rank structure in the design / sample covariance matrix). This is illustrated promptly with a few simple and intuitive synthetic experiments.
2. Large-d-small-n cases are handled by a robust PCR variant under an elliptical family of densities model, that specialize in capturing heavy and dependent tails in the data.

The simplicity of the proposed algorithm is salient:
- Project data on the sparse principal eigenvector of the sample Kendall's tau (akin to sparse PCA on the sample covariance, via the truncated power algorithm).
- Regress Y on Xu.

Other notes:
- line 373, F distribution -> exponential distribution
- why do you scale the prediction error by 100 times instead of scaling the error axis? I might have misunderstood here.
Summary: After rigorously showing clear advantages of PCR vs least squares, the paper presents a novel semiparametric approach on sparse and robust PCR.

I've read the author's rebuttal.

Submitted by Assigned_Reviewer_8

Response to author feedback:
Thank you for clarifying the novelty of the robust PCA approached; it is a good idea to also describe the novelty saliently in the paper. I suggest also explicitly mentioning the possibility of the generalization to the case with more than one PCA component, even if the full proof would not fit in this paper. Simply saying in the Introduction that you consider here the special case of one component, since it is already an interesting case backed up with positive empirical experiments, would already help a lot.


Summary:

The paper generalizes principal component regression (PCR) by providing a robust
variant of the technique based on multivariate Kendall's tau in place of the covariance
matrix. The authors then provide a simple algorithm for learning the model
and, more importantly, analyze its theoretical properties, providing the rate
of convergence. Finally, the model is demonstrated on simulated and equity data.

Quality:

The method presented in the paper is fairly straightforward,
consisting merely of estimating the population Kendall's tau,
computing its first eigenvector, and then performing linear
regression to the observations. The method is very intuitive;
it is easy to understand why this should be preferred to
classical PCR when working with data that has outliers.

The use of Kendall's tau in place of the covariance matrix in
PCA is a nice idea and also well justified based on Oja's
theorem. However, it remains unclear whether this should be
considered as a novel contribution of the paper; no citations
for earlier works are given, but the authors do not seem to
describe it as a key contribution of the paper either. I
believe it has potential for wider impact than the particular
example considered in this paper, and one could imagine
already a paper studying such a robust PCA estimator as a
valuable contribution. To my knowledge, the closest work here
would be the very recent paper by Han and Liu in ICML'13. Can
the authors clarify the relationship with that paper?

The analysis of the theoretical properties of PCR is
valuable. However, the whole discussion is limited to the
special case of simple PCR where the outcome is assumed to be
related only to the first principal component. Do any of the
results generalize to the more general case where the outcome
is related to the first K components? At least the authors
should explicitly mention that they limit the theoretical
analysis to this special case; now the paper never even
mentions the PCR setup I would consider as the standard one.

The experiments seem to be conducted properly and they clearly
illustrate the advantage of the robust variant; such a set is
sufficient for a theoretical paper. I like the fact that the
authors show how the equity data is not normally distributed,
motivating the example.

Clarity:

The paper is reasonably well written, to the degree that a
theory-heavy paper can be. However, the exact scope and
contributions are a bit vague; it is unclear whether the use
of Kendall's tau for robust PCA is novel, the authors do not
mention they limit the analysis to a special case of PCR, and
some of the earlier results are listed as "well known" without
citations.

One problem is that the proofs for the perhaps main
contributions of the paper, Theorems 3.2 and 3.3, are left for
Supplementary material, without even mentioning it in the
paper itself. It is understandable that writing out the full
proofs would take too much space here, but some sort of an
outline would be useful.

Originality:

The paper has two novelties: it presents a novel robust PCR
technique and it provides new theoretical analysis of PCR in
general. The significance of the first relies fully on whether
the use of Kendall's tau in PCA should be considered as a
novel contribution; if not, the algorithm for robust PCR is
trivial.

The theoretical analysis provides new useful results and is
based on very recent works by Oja, Vu and Lei, and Ravikumar
et al.

Significance:

The robust method is clearly useful for anyone applying PCR,
especially in light of the theoretical analysis for the
convergence rate. However, the significance may be limited due
to the fact that both the theoretical analysis and the
simulation experiment rely on the simple model where the
output only depends on the first principal component. The
application to the equity data suggests the method works well
in practice, but is merely one example.


Detailed comments:

- The 2nd sentence of Abstract is very complex and tries to lay out all the dimensions
of the paper at once. You should consider splitting it into at least two sentences.

- Page 3: "As a well-known result..." would need a citation. In general, "well known"
is not good use of scientific language; you use it twice in the same paragraph for
results that are not obvious for most readers.

- Section 3.1: It is good to remind the readers about the elliptical distribution,
but listing all the equivalent formulations is not be necessary when the rest
of the analysis only uses one of them.

Summary: The paper presents a new method for robust principal component
regression and proves interesting theoretical results for PCR in general.
The main shortcoming is limiting the analysis to the simplest case of PCR
with only one principal component.

Submitted by Assigned_Reviewer_9

Summary:

The paper is primarily divided into two parts. The authors first discuss the advantages of principal component regression (PCR) over classical linear regression. After giving an overview of the problem, they provide new theoretical results explicitly showing how principal component regression is insensitive to collinearity, and how it can take advantage of low rank structure in the covariance matrix. These results are taken in the setting where both dimension d and sample size n can increase. The second part of the paper develops a new PCR algorithm to handle the case where d > n, and when the predictors X are from an elliptical distribution. The new method is straightforward: the use of Kendall's tau in place of the sample covariance matrix handles the generalization to elliptical distributions (utilizing recent work by Oja), while the sparsity constraint handles the setting where d > n. The authors then confirm the advantages of the new method in both simulated and real-world data.

Quality:

The primary theoretical results (theorem 2.2 and 3.3) are strong contributions and technically sound. The simulated and experimental results, seen in figures 1, 2 and 3, add significantly to the paper's strength.

One primary quibble is the assumption of the principal component model (equation 2.1) in the first part of the paper, and the assumption of equation 3.5 in the second part. Discussions from Artemiou and Li (2009) and Cook (2007) -- both referenced in this paper -- focus on the advantages of PCR over linear regression in a much more general context. In the current paper, the regression coefficient is explicitly assumed to be aligned with the first principal component, which (based on the aforementioned references) does not characterize all scenarios where PCR outperforms LR. It is thus unclear to what extent the results in figure 1 and 2 are trivial -- is PCR outperforming LR merely because a principal component model was assumed?

The application to real-world data is a strong point of the paper, and the positive result seen in figure 3 helps address the above concern. However, the dataset they chose is one example, and it is unclear whether the strength of RPCR depends on analysis choices, such as the authors' choice to focus on the financial subcategory in their dataset.

Clarity:

The paper is well-written. The organization structure is exceptional, making the paper easy to read. The presentations of the main theorems (2.2 and 3.3) are less clear, understandably due to the fact that their proofs are delegated to the supplementary materials. Some choices, such as the supposition in theorem 2.2 (r*(Sigma)logd/n = o(1)), or the conditions of theorem 3.2/3.3, are not made clear or are not self-evident. Some exposition of the theorems themselves -- and not just their consequences -- would be helpful.

Originality:

The originality of the paper is largely tied to the two main theoretical contributions, theorem 2.2 and theorem 3.3. The development of the RPCR algorithm builds heavily from recent results; however, the synthesis of these results into a novel algorithm contributes to the overall originality of the paper.

Significance:

The two main theorems are significant contributions to the field. As the authors mentioned, theorem 2.2 is the first time observations in PCR have been explicitly characterized. The new method, RPCR, also shows promise to be used by others.

Other notes:
- Line 43: R^{nxd} should be R^{dxd}
- Line 358: The first w_d should be w_3
- Line 361: I could not find a definition for m, but I assume m = 2 in this context.
- There is limited discussion of the constant alpha. For example, LR outperforms PCR for large alpha. Secondly, was there a reason, aside from simplicity, to set alpha = 1 in the simulation studies?
Summary: This paper signifies important contributions to our understanding of principal component regression, and is well-presented. Results would be strengthened by a clearer justification of the choice to assume a principal component model, and by a more thorough analysis of real-world datasets.
Author Feedback

Author rebuttal: With regard to model generalization, we would like to point out that the current proof techniques, with proper modifications, can be generalized to the settings where there are more than one principal component in the regression model (the rate of convergence shown in Theorem 3.3 will be changed to \sqrt{ks \log(kd)/n}, where k is the number of principal components in the model). In the mean time, as pointed out by other reviewers, Theorem 2.2 has been very helpful in illustrating some advantages of PCR over least square regression.